# Unmet Needs and Resilience: The Case of Vulnerable and Marginalized Populations in Nairobi's Informal Settlements

**Ivy Chumo** [1,*] **, Caroline Kabaria** [1] **, Alex Shankland** [2] **and Blessing Mberu** [1]

1   African Population and Health Research Center (APHRC), APHRC Campus, Kitisuru, Nairobi P.O. Box 10787-00100, Kenya
2   Institute of Development Studies (IDS), Brighton BN1 9RE, UK
*   Correspondence: ivychumo@gmail.com

**Abstract:** Catalyzing change and promoting sustainable cities in informal settlements and their residents requires an understanding of unmet needs and resilience among marginalized and vulnerable groups (MVGs). This is because needs identified on behalf of MVGs as "unmet" are sometimes not perceived as unmet, or even "meetable", and resilience strategies from above are often perceived as unsuitable by the MVGs. To the best of our knowledge, no study has used governance diaries to identify the unmet needs and resilience strategies of MVGs from their perspectives. As such, this study explored the unmet needs and resilience strategies of MVGs in informal settlements using governance diaries. This was a qualitative study using governance diaries with 24 participants from two informal settlements in Nairobi, Kenya. We used Maslow's hierarchy of needs for the framework analysis. We identified unmet needs related to physiology, safety, love and belonging, and self-esteem, in the order of the hierarchy. MVGs did not need the full satisfaction of a lower need to yearn for a higher one, and continue living despite their unmet needs. However, there were no self-actualization needs as the participants could not satisfy the lower level needs. The urban paradox reminds us that cities are not always beneficial for all. There is a continued need for holistic approaches to uncover the often hidden resilience strategies for achieving unmet needs. Our study identified behavioural and cognitive resilience strategies. As such, actors need to embrace and build on local resilience strategies in efforts to address the unmet needs of MVGs in pursuit of inclusive urbanization in Africa. The identification of unmet needs and resilience strategies adds to the literature, policy and practice on how and why residents and MVGs continue working and living in informal settlements despite a lack of or inadequate basic amenities. Our study findings imply that actors in informal settlements need to build on and re-build local resilience strategies in pursuit of inclusive and liveable urbanization in Africa, as unmet needs tend to increase with worsened marginality and vulnerability status. Beyond the resilience strategies adopted by MVGs, governments, service providers and caregivers should take more useful actions to prevent or reduce unmet needs.

**Keywords:** needs; unmet needs; resilience; marginalised and vulnerable groups (MVGs); informal settlements; governance diaries

## 1. Introduction

Cities and urban development are complex, with consequences addressed in the United Nations 2030 Agenda for Sustainable Development [1]. Sustainable Development Goal (SDG) 11 intends to "make cities and human settlements inclusive, safe, resilient and sustainable" [2], which correlates with SDG goal 3, which intends to "ensure healthy lives and promote well-being for all at all ages" [2]. Despite the goals, indicators and correlations, there is an increased discrepancy between needs and resources required to meet the raised expectations of the urban poor in low- and middle-income countries (LMIC) [3]. Human needs are a conceptualization of health and wellbeing conditions that must be satisfied for people to stay physically, socially and mentally healthy both individually and on a

societal level [4]. The needs are deemed necessary by an individual or a group [5]. Needs are considered to be "unmet" when support and services are not available [6]. The concept of unmet needs has been distorted and misunderstood as needs are perceived to be ever-changing [7]. Yet, increasing trends of unmet needs in informal settlements is a serious humanitarian crisis which different countries' planners and policymakers are struggling to overcome [3]. In fact, informal settlements witness deprivations of human needs in many cities in LMICs [8,9]. The deprivations and expansion of informal settlements place enormous stress on already struggling systems and on the most marginalized and vulnerable groups (MVGs) who cannot switch to better, private services when in dire need [10]. Kenya's informal settlements do not have legal recognition because they are built in unauthorized areas, mostly with health and wellbeing services which are illegal [8,11] and, onsite, not suitable for achieving SDG 3 [11,12]. Due to illegalities, residents in informal settlements have inadequate or no access to basic services such as the provision of water, sanitation, waste management, health, education, electricity, roads, walkways and lighting [13,14]. The lack of access to basic amenities is correlated with a high level of interpersonal crimes such as domestic violence, abuse, neglect and various types of social ills, with the MVGs at an increased disadvantage, hence exacerbating efforts to satisfy unmet needs and promote sustainable cities [15,16].

Living in informal settlements disproportionately affects certain groups [17,18]. Notably, older persons, child-headed households (CHHs), and persons with disability (PWD) are MVGs because they are disproportionally represented in informal settlements [19,20]. Three elements of CHH vulnerability are (a) biological and physical needs (i.e., inability to act and think like adults); (b) strategic needs (i.e., children's limited levels of autonomy and dependence on adults); and (c) institutional invisibility and lack of voice in policy agendas [21]. Older people's vulnerabilities include lack of access to regular income, work and health care; declining physical and mental capacities; and dependency within the household [22]. Without an income or work, older people are dependent on their caregivers, who also have unmet needs [23,24]. PWD are abused and often (sometimes incorrectly) assumed to be unable to work, hence increasing their inability to meet their needs [25,26]. PWD also have higher rates of poverty and face physical barriers, communication barriers, attitudinal barriers, and a societal-wide lack of sensitivity or awareness [27]. With a clear understanding of different MVGs, it is widely acknowledged that MVGs have additional unique needs and are affected more, both in the short and long term, when their basic needs are not met or are delayed [21,28].

Older persons, PWD and CHHs' concepts of unmet need reflect those of other MVGs in society and are contingent on socio-demographic and contextual factors [6,29]. In cases where the unmet needs of different MVGs are looked into separately, it becomes challenging for actors and stakeholders to address these needs using interdisciplinary approaches [26,30,31]. Therefore, it is important to provide evidence of the unmet needs of multiple groups for insight into areas of common support by any interdisciplinary team [32,33]. Unmet need is a complex concept, with different interpretations according to the perspective taken [31,33,34]. Needs identified by others on behalf of older persons, PWD and CHHs as "unmet" are sometimes not perceived as unmet, or even "meetable" [3,34]. As such, it is important to involve MVGs to define and classify unmet needs. In addition to understanding unmet needs, our study will uncover the resilience of MVGs to unmet needs. Resilience is the ability of MVGs to resist, absorb, adapt, respond and recover positively, efficiently and effectively when they have unmet needs while maintaining an acceptable level of functioning [35]. Harvesting rich local intelligence from the marginalized and the vulnerable is always missing, yet has become more relevant in the context of advancing sustainable cities, inclusivity and participation for the co-production and co-creation of knowledge and solutions. Other studies have shown how accounting for resilience from below can limit negative effects, enhance positive outcomes and act as reference for best practices, yet less has been documented [36]. Moreover, identified resilience strategies on behalf of MVGs are often perceived as unsuitable by the MVGs. As such, our research

questions included, (a) what are the unmet needs of MVGs in informal settlements? and (b) what are the resilience strategies embraced by MVGs in informal settlements? To the best of our knowledge, no study has used governance diaries to identify the unmet needs and resilience strategies of MVGs from their perspectives. As such, this study explored the unmet needs and resilience strategies of MVGs in informal settlements using governance diaries. Our novelty involved the community-based participatory approach of using governance diaries, engaging the community as researchers and having MVGs explore their unmet needs and resilience strategies.

The structure of the paper includes the introduction and literature review with a research gap and research questions described above. This will be followed by the conceptual framework; methods and materials; results; discussions and conclusion.

### 1.1. Literature Review

This section presents the literature of informal settlements in Kenya, marginalised and vulnerable populations, and human needs.

### 1.1.1. Informal Settlements in Kenya

Informal settlements are unplanned sites that are not compliant with authorized regulations [37]. The widespread growth of informal settlements in urban centres in Kenya has become a central debate in urbanisation during the last two decades [38]. Yet, the hesitancy of the Kenyan government regarding improving informal settlements and at least providing the minimum support for basic requirements and services has led to unimaginable suffering among residents [39]. This is coupled with the fact that the government has had a history of failing to recognize the growth and proliferation of informal settlements and, thus, excludes the urban poor from the rest of the city's development plan [8,39]. While constitutional and attitudinal changes are observable, it is hoped that advocating for the urban poor, particularly marginalised and vulnerable groups, would help change the course of events in informal settlements in Nairobi, Kenya.

### 1.1.2. Vulnerable and Marginalised Population

Vulnerability refers to the conditions determined by physical, social, economic and environmental factors or processes, which increase the susceptibility of an individual or community to the impact of hazards [32,40]. A vulnerable group is, therefore, a population that has some specific characteristics that make it at higher risk of falling into poverty [40], those who by virtue of gender, ethnicity, age, physical or mental disability, economic disadvantage, or social status may be more adversely affected in a community than others and may be limited in their ability to claim or take advantage of assistance and related development benefits [22,40]. Marginalization generally describes the overt actions or tendencies of human societies whereby those perceived as being without desirability or function are removed or excluded (i.e., are "marginalized") from the prevalent systems of protection and integration, so limiting their opportunities and means for survival [40]. A vulnerable and marginalized individual/group is defined as a group that, in a particular context because of its relatively small population or for any other reason, is unable to fully access basic amenities and participate in the integrated social and economic life as a whole [40,41].

### 1.1.3. Human Needs

Basic human needs, emotions and capacities are not evil or good; they are neutral. Therefore, needs should be encouraged and permitted to guide our lives, resulting in happiness and growth. If this essential core is suppressed, it may cause sickness and unhappiness [42]. Needs are considered to be "unmet" when support and services are not received [6].

### 1.1.4. Conceptual Framework

Human needs are insatiable and some needs are more important than others. As such, this study is grounded in Maslow's hierarchy of needs. Research related to needs originates from the motivational hierarchy of human needs developed by Maslow [43]. Maslow points out that people satisfy certain needs over others and that once the most basic needs related to biology and survival are met to a certain extent, the necessity of satisfying psychological and bonus/self-fullfillment needs emerges [44], as shown in the "Hierarchy of Needs Pyramid", in Figure 1, where needs are in order of the most pressing.

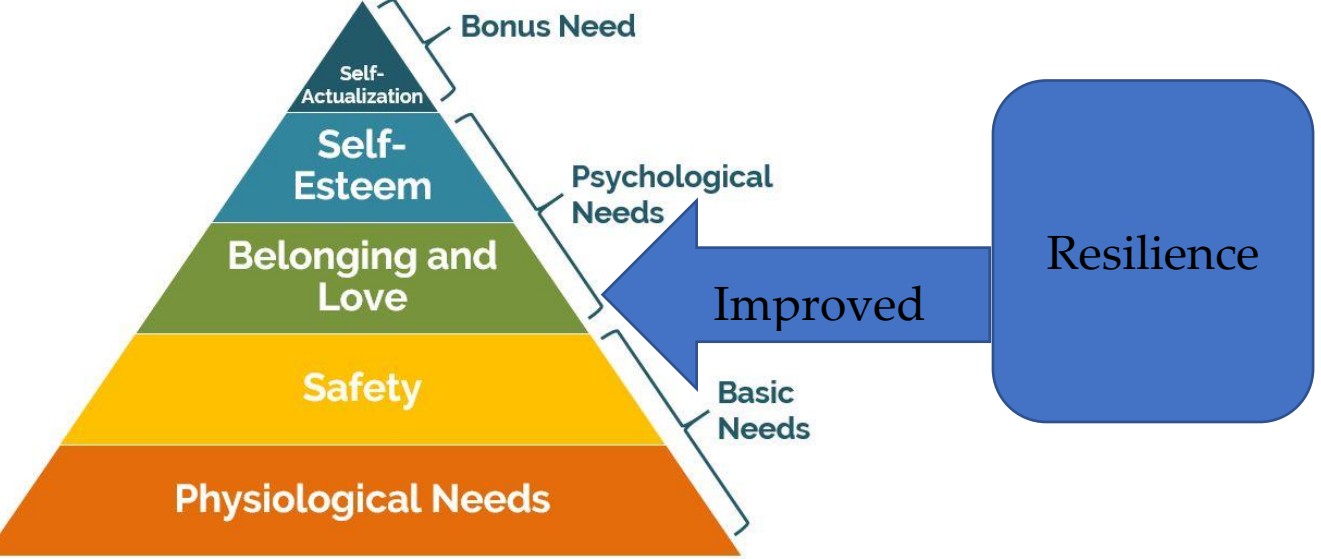

**Figure 1.** Conceptual framework (adapted from Maslow's hierarchy of needs).

According to Maslow's framework, human behaviours or practiced lifestyles are essentially motivated by a desire to satisfy personal needs [43]. The top higher level need is always impossible to achieve when lower level needs are not met to some extent [44]. The theory is applicable to this study because MVGs have biological, psychological and social aspects of their needs that are conditioned to be met in a hierarchical order, progressing from basic/physiological needs toward social and self-actualization needs at a higher level. In informal settlements, due to the many needs and the few resources to satisfy them, behavioural tendencies are tilted towards the negative (i.e., crime and environmental degradation) and higher order needs are rarely met. The effects of unmet needs are felt by most MVGs and many of them have adopted resilience strategies. Resilience strategies enable MVGs to be in a condition wherein a lower level need has been partially met and they can then proceed to seek a higher level need. Resilience increases the gratification of needs (goals), increased motivation, and a desire for more, resulting in growth and more satisfaction.

## 2. Materials and Methods

The study is reported per a set of standardized criteria for reporting qualitative research (COREQ) [45].

### 2.1. Aim/Objectives

Our study sought to explore the unmet needs and resilience of MVGs in informal settlements using a governance-diaries approach. This study is part of a greater ongoing work by a multidisciplinary multi-country and multi-year study under the Accountability and Responsiveness in Informal Settlements for Equity (ARISE Hub). The Hub is a research consortium, set up to enhance accountability and improve the health and wellbeing of marginalised populations living in informal urban settlements in low- and middle-income

countries with partners in Bangladesh, India, Kenya and Sierra Leone, with great emphasis on the use of quantitative and qualitative mixed-methods research, especially community-based participatory-research (CBPR) approaches.

### 2.2. Study Design

This was a qualitative study performed using the governance-diaries method. Governance diaries are an ethnographic approach using more than one method of data collection and where participants make regular records of their daily activities and experiences [46–48]. For this study, governance diaries included in-depth interviews (IDIs), which were complemented by participant diaries, informal discussions, participant observations and reflections. Governance diaries are typically used in contexts where there is a need to explore the depth of everyday life, as time allows researchers to spend longer periods in the field for exploration, unlike other qualitative studies, where data collection is collected once from study participants [46,49]. For our study, the researchers got the opportunity to explore unmet needs and resilience strategies in depth by spending 4 months with the study participants.

### 2.3. Study Setting

The study was conducted in Korogocho and Viwandani informal settlements in Nairobi, in the areas covered by Nairobi Urban Health and Demographic Surveillance System (NUHDSS) initiated in 2002 by the African Population and Health Research Center (APHRC) [50]. As of 2019, the number of household structures in Korogocho and Viwandani were 31,961 and 56,837, respectively, with an average of five people in every household. Korogocho has a stable and settled population and residents have lived in the area for many years [51], while Viwandani is located next to an industrial area with many highly mobile residents who work or are seeking jobs in the industrial area [51]. MVGs identified through a social mapping process included persons with disability (PWD), older persons and child-headed households (CHH). Further, each of the informal settlements has 8 villages, which acted as a guide during the selection of study participants (Figure 2).

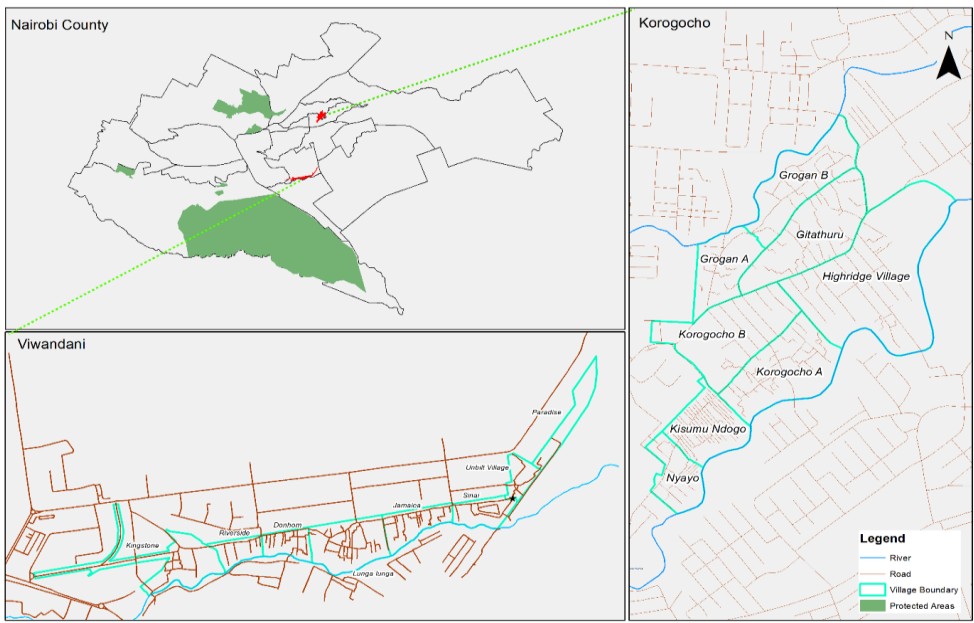

**Figure 2.** Study sites.

### 2.4. Target Population, Sampling and Sample Size

The population of interest was PWDs, CHHs and older persons (i.e., groups who were identified at the social mapping phase). We selected 24 participants comprising 4 PWD, 4 CHH and 4 older persons in each of the study sites. The participants, who were residents and benefiting from services from at least two villages, were purposely selected to represent

the villages in the communities if they had resided in community longer than others who were identified in the category. The longer period of stay in a community meant that they were knowledgeable about issues in the community.

*2.5. Data Collection Process*

Community advisory committees (CACs) (i.e., individuals selected by the community to represent and act as a liaison between the researchers and the community), co-researchers (i.e., community members recruited as research assistants because they have a better understanding of the context and closer rapport with respondents) and researchers worked together in the recruitment of study participants. We used governance diaries to collect data from January to April 2021 on questions related to priority needs and resilience strategies of MVGs. Diaries approach entailed IDIs, participant observation, participant diaries, reflection and informal discussions. IDIs were the dominant method and were informed by the other methods. Below is the description of the data-collection process:

Observations: These included observation by the co-researchers, which allowed for a holistic awareness of events as they unfolded and, as such, enabled a comprehensive understanding of what matters to respondents. We also observed the environment related to our study subjects including observing health and wellbeing services. These observations resulted in photos and insights into what to probe further in the IDIs. The observations were conducted before, during, and after the IDIs to complement the discussions recorded. Reflexive discussions informed the content and concepts for observations.

Reflexive discussions: Reflective discussions were held between pairs of co-researchers daily, among the whole group of co-researchers weekly, and between researchers and co-researchers every two weeks, to understand the outcome and determine emerging themes and gaps to be probed during subsequent IDIs and routine observations.

Participant diaries: We provided the study participants with a diary guide on daily activities related to unmet needs and resilience pasted on the front of a diary. Each participant wrote about daily activities related to unmet needs and resilience strategies, without writing their names. Co-researchers occasionally called the participants and conducted impromptu visits to remind participants about diary-writing activities.

Informal discussions: An informal conversation was carried out between the participants and the researchers to find out key insights and to create rapport with the study participants before the IDIs. The discussions were incorporated into the IDIs.

IDIs: We used study guides with questions on unmet needs and resilience strategies concerning the MVGs. IDIs for subsequent visits on the same questions were adapted based on observations, reflections, informal discussion and participant diaries (Figure 3). In-depth discussions between the co-researchers and the study participants were administered in pairs of 2 co-researchers: one who was moderating the interviews and the second acting as an observer, note taker and facilitator of the recording of the conversations. We reached saturation in the IDIs during the sixth visit when we were approaching the fourth month.

The outputs from informal discussions, observations, participant diaries and reflexive discussions informed and enhanced robust probing during IDIs. For example, if the co-researchers observed an ambulance or burst water pipe, during IDIs, they would probe more on the pipe bursts and the ambulance. The multi-pronged ethnographic data-collection processes are summarized in Figure 3 below:

Co-researchers received training for 6 days on the aims of the study, data-collection process, data-collection tools, and research ethics. We piloted the Swahili-translated study tools with one older person, PWD and CHHs in each of the study sites, followed by a debriefing to assess if the study approach and study tools were well-understood by both co-researchers and study participants. The pilot exercise also enabled us to adjust the translated guides to concepts understood by the study participants and to estimate the time an interview could take. We excluded participants in the pilot from the main study.

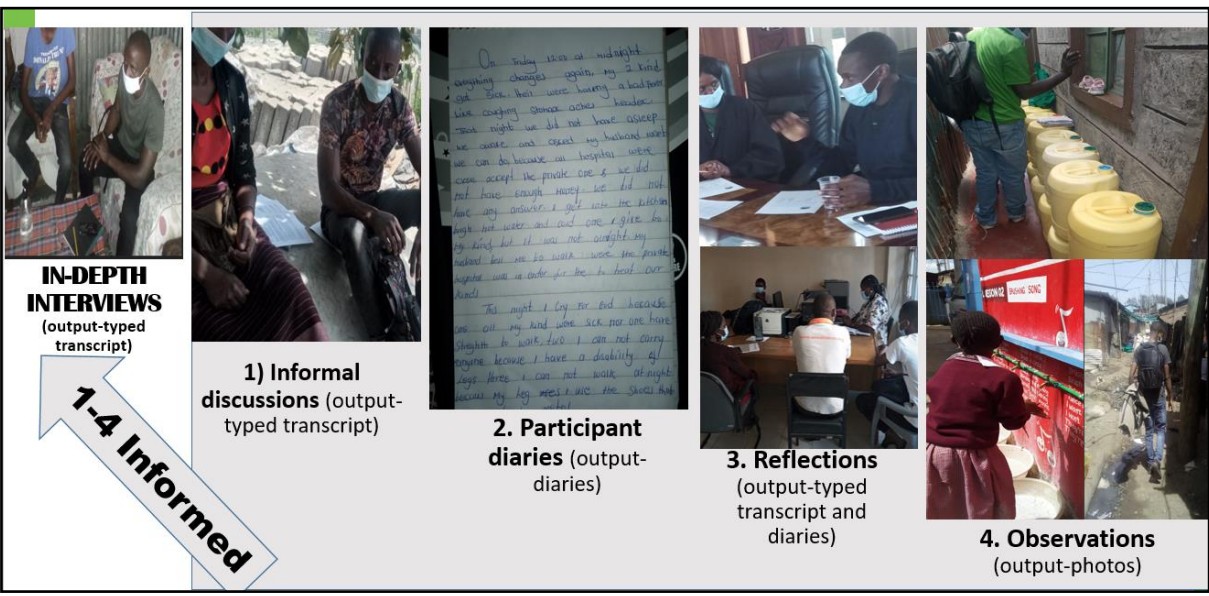

**Figure 3.** Data-collection process.

*2.6. Data Management*

Recorded audios from IDIs, reflections and informal discussions were translated and transcribed from Swahili to English for validity and saved as individual Microsoft Word documents. Outputs were assigned number codes to prepare for analysis and to ensure confidentiality. Outputs that could not be typed (photos from observations and participant diaries) were scanned and saved in a safe folder, as they were reference materials during data analysis. Daily data reviews were convened by researchers and co-researchers to check data accuracy and data completeness.

*2.7. Data Analysis*

Transcripts were imported into NVivo 12 software (QSR International, Australia) for coding and analysis. NVivo is a qualitative-data-management software which can be shared and worked on in groups and facilitates thinking, linking, writing, modelling and graphing in ways that go beyond a simple dependence on coding [52]. We used a framework analysis [53] informed by the Maslow's hierarchy of needs [43]. Framework analysis is adopted for research that has specific questions, a pre-designed sample and pre-existing issues [53]. The first step of framework analysis was listening to the recordings to familiarize the researchers with the information related to unmet needs and resilience strategies. To ensure reliability, two researchers (an experienced qualitative researcher and an anthropologist) and 5 co-researchers, who collected the data, participated in the development of a coding framework by reading the outputs imported into NVivo 12 software, participant diaries and photos independently to establish an inter-coder agreement. Once the initial coding framework was completed, the team met to discuss the themes generated and to reach an agreement on themes (Table 1). The two researchers proceeded with coding, charting, mapping and interpretation of transcripts.

**Table 1.** Themes for Analysis.

| Major Themes | Sub-Themes |
|---|---|
| 1. Unmet needs | a.  Physiological needs<br>b.  Safety needs<br>c.  Love and belonging needs<br>d.  Self-esteem needs<br>e.  Self-actualization needs |
| 2. Resilience strategies | (a) Cognitive strategies<br>(b) Behavioural strategies |

*2.8. Ethical Considerations*

The data collection was conducted in privacy at the convenience of the study participants and the co-researchers ensured that no interruptions occurred and that only authorized study participants listened during the interviewing process. During data collection, each participant was assigned a unique identification code which was used in place of his or her name, allowing for anonymity. Identifiers for all the study participants in this study are only accessible to project staff. All participants provided informed consent prior to data collection and received a copy of the consent form. If the participant verbally consented to participate, a paper consent was signed by the study participants and approved by the co-researcher. Participants were made aware of their right to skip any question and end interviews at any time. Distressed study participants were referred to relevant institutions for counselling and referral for specialized services, as co-researchers received sensitivity training on how to identify distressed participants and how to refer them to relevant community institutions for care. Throughout the data collection, researchers and co-researchers endeavoured to respect the culture, values, and beliefs of the study participants by valuing and respecting the views and opinions of the surrounding communities and the study participant. The study was approved by AMREF Health Africa's Ethics & Scientific Review Committee (ESRC), REF: AMREF-ESRC P747/2020. We also obtained approvals from National Commission for Science, Technology and Innovation (NACOSTI), REF: NACOSTI/P/20/7726. Approval was also obtained from the Liverpool School of Tropical Medicine (LSTM) and the African Population and Health Research Centre (APHRC) internal ethical review committee. All participants provided informed written consent before participating in an interview, including consent for using photos and videos if there were any.

**3. Results**

A first glance at the concept could indicate that "unmet need" implies that there exists no action whatsoever to meet a need for sustainability, healthy and liveable cities by MVGs. From our study, this is not necessarily the case. There could be an action or a strategy, which is deemed inadequate for sustainability, healthy and liveability of MVGs in informal settlements. As such, we explored the unmet needs and resilience strategies of MVGs in informal settlements using a governance-diaries approach by asking two questions (1) what are the unmet needs of MVGs? and (2) what are the resilience strategies embraced by MVGs in two informal settlements in Nairobi, Kenya? presented below. A broad range of issues were reported; while some were described as direct needs, others were depicted as the impact of unmet needs.

*3.1. Theme One: Unmet and Priority Needs*

3.1.1. Physiological Needs

Physiological needs are the needs that have to be satisfied for the continuation of an individual's biological structures [44]. Study participants described that when basic needs were not met to a certain extent, then higher order needs were unmet. The physiological needs of older persons, PWD and CHHs identified as unmet priority needs included water,

sanitation, hygiene, health, education and solid-waste management. The specific needs were prioritized by the different MVGs for the different study sites. Needs for health, sanitation, hygiene and water were common for all MVGs in both informal settlements. Education needs were common to all MVGs with the exclusion of CHHs from one informal settlements who did not feel the impact of education as they relied on solid-waste management (SWM) work. SWM was a felt need in an informal settlement which was further from the dumping site (Table 2).

**Table 2.** Priority and unmet physiological needs.

| Study Group | Viwandani-Priorities | Korogocho-Priorities |
| --- | --- | --- |
| Older persons | Solid-waste management, sanitation, health, education and water | Education, health, hygiene, sanitation and water |
| Child head of households | Education, water, sanitation, hygiene, solid-waste management and health | Health, hygiene, water and sanitation |
| Persons with disability (PWD) | Solid-waste management, sanitation, health, water, hygiene, and education | Education, water, health, hygiene and sanitation |

Water Needs

The need to access water was unanimously pronounced, yet, the study participants did not comprehend the reasons for water shortages. An insistent interplay between water needs, water usage challenges and the vulnerability of study participants emerged in most interviews despite not being directly queried. This highlights the reluctance and helplessness of the MVGs as they described how the situation was persistent, yet they could not do anything about it. As such, access-to-water challenges were mostly felt by MVGs compared to non-MVGs.

> *"We mostly lack water . . . we do not understand the reason for water shortage and as such we no longer use water as we need compared to others"* (IDI, Male CHH, Korogocho).
> *"I told you even in your last visit that there is no water . . . don't ask me this again"* (IDI, Female PWD, Viwandani).

Health-Care and Medical-Service Needs

Three primary health needs were identified across the three groups: (a) mental ill-ness, (b) food-and-nutrition-related health conditions, and (c) chronic physical health and injuries. The health needs were commonly attributed to stress and anxiety due to inadequate resources, lack of transportation to access free health services and food relief, accidents, age, disability status and common illness. Study participants frequently discussed how the needs were not satisfied in many cases because of a lack of finance, as stated by female participants, and in some cases because of ignorance of where to seek help, as stated by male participants. The unmet health needs mean that MVGs did not have control over their health.

> *"I did not have money so after diagnosis, I just stay without taking drugs"* (IDI, Female PWD, Korogocho). *"I feel pain and do not know where to go for the pain to end"* (IDI, Male older person, Korogocho).

Solid-Waste Management Needs

Solid waste was poorly managed in the community and there was a need for effective solid-waste management. The status of solid-waste management was described to have worsened at the time of the study compared to the past. As study participants were mostly concerned about how it could affect hygiene and health in the community.

> *"Garbage is all over in this community . . . Did you see some piles of garbage along the way as you were coming here, it affects the health and hygiene here? Those were better sometimes back"* (IDI, Female CHH, Viwandani).

Sanitation and Hygiene Needs

There were sanitation and hygiene needs linked to cost, distance and availability. The cost that was thought by service providers to be affordable to all, was not always affordable to the MVGs. The distance and availability of sanitation and hygiene facilities expected by service providers to be appropriate, was not always appropriate as older persons, PWD and CHHs in many instances faced exceptional challenges due to their conditions. Older persons and PWD, in many instances, had unique sanitation and hygiene needs related to design that were not always addressed by service providers.

*"Toilets are there but you must pay five shillings, but many times we do not have money"* (IDI, Male PWD, Viwandani). *"Hygiene is a problem because we do not have enough hand washing stations. The hand washing stations are also not designed well"* (IDI, Male Older person, Korogocho).

Education Needs

Education needs not only entails numeracy and literacy or a lack of it. Despite basic education in Kenya being described as free from any expenses, MVGs could not afford school uniforms or other related expenses. The lack of uniform implies that the MVGs could not learn or their children could not learn adequately, leading to poor performance and, ultimately, dropping out.

*"I have no uniform or the books, this makes me miss school sometimes and such things prevent me from going to school. That's why some children drop-out from school"* (IDI, Male CHH, Viwandani). *"My child was harassed because she did not have school uniform ... the other children made fun of her and she is no longer going to school"* (IDI, Male PWD, Korogocho).

### 3.1.2. Safety Needs

When physiological needs are relatively satisfied, new needs arise which are classified as safety needs [44]. These are the needs such as protection from danger, being confident, not feeling fear and being secure economically [34]. Our study participants pointed out the safety needs of security, non-harassment, and freedom from stress, as they relate to the water, sanitation, hygiene, education, solid-waste management and health services of the vulnerable populations. The safety needs were felt by the MVGs who were often not able to afford to switch to services located at safe spaces. Study participants described how insecurity at the handwashing station, water point, sanitation facility, health facility or at the education facility reduced their ability to use the facilities. It was always the wish of the MVGs to access safe basic amenities in safe spaces. Insecurity meant that MVGs did not enjoy the full benefits of basic amenities and sometimes they would forgo the amenities at the expense of their safety.

*"It is not always safe at the hand washing station. It is not safe as you head to the health facility and the water point ... When it is night, we fear using the toilet. One time a lady was raped. We are always worried about the boys who collect garbage here because they can take things in our household without our knowledge. In many cases we forgo"* (IDI, Female CHH, Viwandani).

### 3.1.3. Belonging and Love Needs

Once the physiological needs and safety needs are relatively met, the need for love, commitment and belongingness emerge [43]. Humans are social beings, and as such individual needs for belongingness and love represents a wide range of needs, such as feelings of belonging (i.e., group membership, clubs, churches, and business associations) and love. MVGs expressed a need for love in their relations with individuals and groups. The participants felt the need to be accepted by others, establish friendships, be with relatives, exhibit love to people around and expect love from them. Participants said that achieving these needs decreased isolation and stigma and, thus, enhanced social networks

and love. Belongingness-support needs achieved included, but were not limited to, support and referrals due to health care needs and the choice of friendly facilities.

> *"I went to our neighbour who gave me drugs he had told me her son used and I also used the same drugs"* (IDI, Male CHH, Korogocho). *"I was sick during the past week and my neighbour referred me to a chemist where I bought drugs at a cheaper cost."* (IDI, Female CHH, Viwandani).

> *"I chose to go to that water point because the group that owns the facility is friendly. They allow you to fetch water on credit . . . Also when I have no one to carry water to my house, they do it for me at no cost."* (IDI, Female PWD, Korogocho).

### 3.1.4. Self-Esteem Needs

Once the love and belonging needs are relatively met, there are two kinds of esteem needs. The first is the need to be appreciated and respected by others related to the reputation of a person, such as status, recognition, and appreciation. The other one is the need for self-appreciation and self-esteem, such as self-confidence, independence, success, and talent [43,44]. According to Maslow (1943), except for a few pathological exceptions, all people have a need or desire for stable self-esteem and self-respect based on true capacity, success, and respect for others [34]. Study participants acknowledged the self-esteem needs for respect, self-recognition, and freedom.

> *"I have not been respected by any individual groups when distributing resources"* (IDI, Female older person, Korogocho).

> *"I do not have the freedom to use public resources like the rest who are not disabled"* (IDI, Male PWD, Viwandani).

There were no self-actualization needs mentioned. Full satisfaction of needs is not required to be able to move from a certain need to a higher one, as people who are satisfied with some upper level needs may sometimes still have lower level needs. However, lower needs ought to be met to a certain extent, for one to develop a higher level need. The presence of unmet needs calls for resilience strategies.

### *3.2. Theme Two: Resilience Strategies*

We asked the study participants about their resilience strategies and they identified and described cognitive and behavioural resilience strategies.

### 3.2.1. Cognitive Strategies

The cognitive coping strategies included positive acceptance of issues, focus on the present or future, including daily tasks, active forgetting of bad experiences, favourable comparisons of their experiences and belief in an internal locus of control.

Most participants described acceptance of the issues identified through a process of adjustment, coming to terms with their unmet needs and a focus on things they can do (rather than those they cannot). They also provided a positive interpretation of the situation, such as overall contentment, and the ability to maintain a good quality of life.

> *"When I do not have the basic amenities, I just got to accept it. I think if things were meant to happen, then they will happen"* (IDI, Male CHH, Korogocho).

> *"As my age advanced, my life changed obviously, but I think I'm adjusting and doing what I can do best like chopping vegetables and other household chores and seeking support on things that I cannot do on my own. I cannot go to the hospital or fetch water on my own. We have to accept the situation and find a way to deal with this new life"* (IDI, Female older person, Viwandani).

Another form of adaptive thinking by our participants is a focus on the present and/or the future. This often entails an acceptance of past experiences and hope for the future, expressed as confidence and curiosity regarding things to come.

*"I do what I can to access water and be healthy. When there are health camps, I look for someone to go with me. I prefer that rather than looking back at what I have lost and crying over them"* (IDI, Male PWD, Korogocho).

*"I love learning new things and networking with people despite my age. In doing that, I connect with people who empower me and assist me in meeting some of the basic needs"* (IDI, Male CHH, Viwandani).

In addition to focusing on the present or the future, some of our participants made efforts to actively forget past adverse experiences to protect their mental health. This was done by intentionally forgetting or learning to suppress negative experiences as a form of adaptation.

*"I try to forget things that have happened to me for instance when I was verbally abused while waiting to fetch water. Sincerely, I don't think about the bad things that happen to me because it makes me not do my best in finding solutions"* (IDI, Female older person, Korogocho).

Further, participants were content when they viewed their lives and issues relative to other people and other circumstances. Being fortunate was reported, that is, feeling contented about how things are compared to how they could be, for example being creative, and having family nearby. Participants made comparisons to other people, other times, and other outcomes. Favourable comparison motivated the MVGs in challenging situations.

*"I have done very well and I am very pleased. I have been very fortunate, my speech, memory and creativity is ok. It is only that I cannot see but my family is close to me"* (IDI, Male PWD, Viwandani).

*"In this village, when it comes to cleanliness and access to water, we are not doing badly. This is because I do not see much garbage in most paths compared to the neighbouring village and rationing only happens three times a week"* (IDI, Female older person, Korogocho).

Participants also described an awareness of limited resources or rationing, frequently perceiving that people other than themselves will be more in need.

*"I was still receiving some counselling support but when I saw a lady who was crying, I allowed them to get in for the support"* (IDI, Male CHH, Viwandani).

Some participants invoked an internal locus of control and report trying to focus on their sphere of influence to try to self-activate.

*"It was not my choice to have been born with a disability and so not able to go and fetch water on my own or access health care on my own, but it is my choice to find someone to take me to medical health camp when they are brought up by organizations to the community"* (IDI, Female PWD, Viwandani).

### 3.2.2. Behavioural Strategies

Study participants described behavioural strategies which included using work as a distraction, withdrawal from stressors, connecting to cultural roots or faith, seeking mental-health care, developing self-ascribed resilience as an enduring capacity, processing through creative outlets, developing character traits and learning life-long positive attitudes.

Participants described using work as a distraction from negative thoughts and withdrawing from the negative.

*"I try to work, even if it involves volunteering to collect waste or to clean the community so that I do not sit down thinking about how I do not have money to purchase water or to take my child for health care"* (IDI, Male PWD, Korogocho).

Other participants sought out environments that feel familiar and allowed them to connect to cultural roots or faith. The participant spoke about how connecting to culture (i.e., people and religion) inspired them to overcome initial feelings of emptiness. By talking

with others, they ascertained what is likely to happen, and what difficulties are normal for someone at their age or for someone who has a similar condition. Participants gathered and used information to reflect on their own situation. Considering their own lives in the context of culture and others with common characteristics made MVGs feel better about their own situation regarding unmet needs.

> *"When you attend church services it helps- you get connected to free services and find some updated information of support within the community"*(IDI, Male Older person, Viwandani).

Several of our participants had sought mental-health care and other forms of counselling to deal with past and present stressors. Some of their reflections on seeking this type of support showed that overcoming the stigma surrounding mental health and mental-health care is often part of the process. By sharing issues with others, participants asserted that it helped them feel that difficulties are normal for someone at their status, or for someone who is of their age. Participants gathered and used the information to reflect on their situation, considering their own life in the context of others, this made them feel better about their situation.

> *"In some cases when I go through severe abuse, my colleagues connect and direct me for free services, they also console me that I am not the only one"* (IDI, Female CHH, Korogocho).

> *"When you talk to other people who have the same situation as yours, you find that you are better off. Some even do not walk at all and cannot see at all"*(IDI, Male PWD, Viwandani).

Self-ascribed resilience as an enduring capacity was some participants' experience, as they described their strength in facing adversities as a trait or lasting characteristic rooted in their personality.

> *"I have learnt to always be calm and always satisfied with life. There is a time I went to receive some donations for my children, but by the time I got to the distribution point, people were fighting . . . I got satisfied with three pens that one woman shared with me"* (IDI, Male older person, Korogocho).

Some participants presented that their positive character or active outlook enabled them manage their daily lives, difficulties and unmet needs. The desire to present oneself as having pride, independence or determination helps to explain how participants managed their unmet needs. Pride was identified when talking about their ability to cope with unmet needs. Independence (being independent of the help of others) was reported by some who did not want to seek or receive help for needs that they could meet on their own. The desire to present themselves as having pride and independence helps to explain how participants manage their identified issues and needs.

> *"One has to have a will . . . It would be easy I think to sit back and wait for someone to help. You have got to be willing to do what you can . . . I have always been able to use sanitation facilities on my own. I even go and fetch water. It might take long but I get to accomplish"* (IDI, Female older person, Viwandani).

> *"Even if there is no water, it will not go beyond one week"* (IDI, Female PWD, Viwandani).

## 4. Discussion

Our study findings uncovering the unmet needs and resilience strategies of MVGs using governance diaries will contribute to policy, practice and methodology. First, by contributing to the 2030 Agenda for SDG 3 on health and wellbeing, SDG 11 on sustainable cities and the New Urban Agenda opportunities to strengthen the liveability of cities and to shape urbanization to account for the health and wellbeing of MVGs and their social sustainability in cities. Social sustainability has to be considered as a dynamic concept, just like human needs, which will change over time due to external influences, prompted by changes in service delivery by local authorities, hence the need for individual resilience.

Secondly, knowledge of unmet needs and resilience strategies is useful to actors working in informal settlements, as the actors gain clarity on what services are urgent and/or important, and how MVGs act when there is an unmet need. We grounded our work in Abraham Maslow's hierarchy of needs in the exploration of the five levels of needs, where certain basic needs must be relatively met before any consideration is given to higher order needs [43,44]. The findings around physiological and safety needs are profound and indicative of a persistent challenge which is consistent with previous findings across all residents of Nairobi informal settlements during the Nairobi Cross-Sectional Slum Survey (NCSS) [54], as well as a study on causes of adult death using verbal autopsy data from the NUHDSS between 2003 and 2012. In 2002, insecurity was not identified by all slum residents as a first-order challenge but in the 2014 report it was identified as a first-order challenge [50]. The same was the case for garbage disposal [54]. In a 2015 study on trends in causes of adult deaths among the urban poor, using evidence from the NUHDSS, a substantial epidemiological transition was shown, with deaths due to non-communicable causes experiencing a four-fold increase from 5 % in 2003 to 21.3 % in 2012, together with another two-fold increase in deaths due to injuries from 11 % in 2003 to 22 % in 2012. Related to the level of insecurity and safety concerns in the two study communities is evidence that the major causes of injury-related deaths over the study period were assault (54%), road traffic accidents (20%), and exposure to smoke/fire/flame (14%) [55]. It is, therefore, instructive that the same challenge identified among the general population had not been resolved for MVGs several year afterwards, despite their other relative disadvantages. Our findings showed that the relative provision of basic needs form a strong foundation for the need to be loved.

The love and belonging needs always reminded individuals of the almost permanent need for social capital, identity and association. Understandably, no man is an island unto oneself in society [44]. Cultural identity, social grouping, association, affiliation and belonging are greatly needed to enhance harmonious human co-existence in various social settings [56]. Achieving love and belonging lays a basis for self-esteem and self-actualization. The hierarchy of needs implies that when people are guaranteed most basic needs as an individual, community, society or association, they can muster the courage and confidence to contribute to the achievement of higher ordered needs and can reasonably contribute to a liveable and healthy society. MVGs in informal settlements did not suggest specific self-actualization needs, due to vulnerabilities and the difficulties in informal settlements and the inability to fully meet the lower level needs. The challenge of poverty in the slums has been frequently document but the case of Nairobi is particularly challenging and with the urban-poor food insecure spending up to 60 to 65 percent of their income on food, many resort to coping strategies such as restricting consumption, eating fewer or smaller meals and eating cheaper products. In such a context, where only 12 percent of needy households receive food assistance [57], the living conditions of MVGs can only be a matter of utmost concern. Self-actualization is ongoing—growth oriented, it is a pressing forward towards fullness—good values, serenity, kindness, courage, honesty, love, unselfishness and goodness. Our participants were not self-actualized. People who are not self-actualizers are deficiency-motivated and categorize things, people, and events. The result is a valuing, judging, interfering, condemning attitude towards others and life at large. They are need-motivated, which often results in exploiting, blustering, and overriding with a selfish need to control [42]. Methodological contribution can be attained, as it was in this study, where MVGs were involved in participatory approaches to identify their needs and resilience strategies through governance diaries.

Due to the unlimited [34], competitive and recurring nature of needs [56], resilience strategies are important for survival, because in instances where MVGs ought to live and work without meeting basic needs, resilience is important [36]. Results showed that adopting resilience is imperative because informal settlement, as a system, can reboot itself in a degenerative vicious cycle of unmet basic amenities and could make the cities unliveable and unhealthy. As such, study participants used specific cognitive and behavioural coping

strategies and ways of framing circumstances to maintain their functioning and wellbeing despite the unmet needs. The cognitive coping strategies included acceptance, focus on the present or future, active forgetting of bad experiences, favourable comparisons of their experiences and belief in an internal locus of control. Further, behavioural strategies included using work as a distraction, withdrawal from stressors, connecting to cultural roots or faith, seeking mental-health care, developing self-ascribed resilience, processing through creative outlets, developing character traits and learning life-long positive attitudes. A few of these strategies have been identified in other studies as well [36]. For example, displaying acceptance and choosing is an adaptation to unmet needs [58,59]. However, to our knowledge, this is the first study describing unmet needs and resilience strategies from the view of multiple MVGs using a governance-diaries approach. MVGs are usually left out of decisions that involve them; hence, needs and strategies reported usually do not reflect their daily experiences. In instances where a single MVG is involved, few resilience strategies are identified [60], hence limiting the choice of strategies to inform context-specific practice and policy. Cities and subnational governments will need to consider the adoption of a multidimensional approach to addressing unmet needs [14]. Investing in actions for unmet needs is a critical policy lever for addressing the needs of MVGs. This is because people may not identify their issues as needs, if they feel able to manage the issue (i.e., through a level of acceptance or favourable comparison), or if they are unclear or unaware of the need.

The implication of our study is contingent on two factors. First, policy makers and urban planners should know why informal urban residents continue living in informal settlements even when there is the paradox of urban advantage. Second, programs for resilience should be able to reach informal urban residents with unmet needs. To the extent that reasons for unmet needs reflect deficiencies in the service provision and a lack of government support, actors should endeavour to remove those obstacles. In contrast, if an unmet need is primarily rooted in discrimination and current societal roles, improved service delivery alone will not be sufficient. In such cases, an appropriate program response might well involve well-designed communication programs. Communication activities might address these social issues, making people more aware of support needed for MVGs.

*Strengths and Limitations*

While the sample is limited by size and geography, it nonetheless allowed exploratory insight with multiple MVGs using Maslow's hierarchy-of-needs framework to describe the needs of MVGs. Purposive sampling was used as a method to select cases rich in the relevant information. While findings may not be generalisable to individual groups of MVGs at large, they help inform an understanding of the identification of unmet needs and resilience strategies of multiple groups of MVGs. Although interviewee accounts can be seen as a hybrid of facts and fiction, interviewing participants using governance diaries in their homes was a way of capturing relevant and varied dimensions of the accounts. Despite these limitations, our ethnographic CBPR study contributes to a nuanced picture of unmet needs and resilience strategies, as it brings out dimensions that are not easily captured in quantitative studies.

## 5. Conclusions

The urban paradox reminds us that cities are not always beneficial for all. The role of cities in sustainable development has become more prominent, more so with the growing population in informal settlements. As such, the social sustainability of health and wellbeing needs can be harnessed through resilience strategies in informal settlements, where the inequality of MVGs is the order of the day. Marginalized and vulnerable groups often tolerate services designed for a general population, and have many unmet needs. Needs are dynamic and are likely to shift over time. As such, unmet needs are not simply present or absent, but may slowly increase or decrease over time, and be affected by shifting circumstances, strategies and resources. The realization of unmet needs in informal

settlements demands an appropriate application of Abraham Maslow's hierarchy of needs with a specific focus on the most urgent basic needs, which must be achieved before any consideration is given to other human needs. By examining the needs and resilience of MVGs, responsible state actors can help the groups to successfully remain to live viably and independently.

Access to basic amenities can be a matter of life and death for MVGs. Understanding and building on resilience strategies would assist the MVGs to have better liveable cities. The notion of resilience should go beyond resilience, and attempt to change the underlying conditions that create and exacerbate the status of unmet needs. The idea of challenging resilience is to go beyond building back better to building differently in a manner that does not preserve the existing situation, but rather confronts unmet needs. If measures for building back differently are implemented, MVGs will counter negative trends and galvanize actions toward achieving equitable, inclusive and resilient urban futures. Informal settlements cannot be liveable unless there is a collaborative, constructive and skilful engagement, aimed at understanding the unmet needs and resilience of residents and cascading services down to MVGs. Consequently, development and policy actors in government, public and private institutions, workplaces, family, societal and the community should always consider transformative measures to confront the unmet needs of MVGs living and working in informal settlements for healthy societies. This is because informal settlements are facing numerous needs and addressing the needs requires intersectoral collaboration among actors, co-creation of solutions and interventions and building on local knowledge and capacities in informal settlements.

There is a continued need for holistic approaches to uncover the often hidden resilience strategies for achieving unmet needs. As such, the identification of unmet needs and resilience strategies adds to the literature, policy and practice on how and why the urban poor continue working and living in informal settlements despite the lack of or inadequate basic amenities. Unmet needs increase with worsened marginality and vulnerability status and the lower incomes of caregivers. As such, actors should build on local resilience strategies in an effort to address the unmet needs of MVGs in pursuit of inclusive urbanization in Africa. Beyond resilience strategies adopted by MVGs, governments, service providers and caregivers should take more useful actions to prevent or reduce unmet needs. The data collection using governance diaries helped to capture the lived experiences of participants with regard to unmet needs and resilience strategies among the MVGs in informal settlements in Nairobi, Kenya, and lessons may be transferable to other informal settlements, even outside of Kenya. The findings may, however, be different in rural or formal settlements, and further research should explore and clarify this point.

**Author Contributions:** Conceptualization, data curation, formal analysis: I.C. Methodology: I.C., C.K., A.S. and B.M. First draft writing: I.C. Reviews: C.K., A.S. and B.M. All authors have read and agreed to the published version of the manuscript.

**Funding:** GCRF Accountability funded the project activities through UKRI Collective Fund award with award reference ES/S00811X/1.

**Institutional Review Board Statement:** The study was conducted in accordance with the Declaration of Helsinki, and approved by the AMREF Africa Ethics and Scientific Review Committee (ESRC/P747/2019; Date: 8 February 2020) and the National Council for Science, Technology, and Innovation (NACOSTI/P/20/7726; Date: 20 November 2020). We also received broader ethical clearance from the Liverpool School of Tropical Medicine (Protocol: 19-089; Date 21 January 2020).

**Informed Consent Statement:** Informed consent was obtained from all subjects involved in the study.

**Data Availability Statement:** The data presented in this study are available on request from the corresponding author. The data are not publicly available because other manuscripts are still being written using the same data.

**Acknowledgments:** We extend our gratitude to our study participants who agreed to participate in this study. We also thank our research team who supported the whole research process. We

also thank the community leaders in our study settings. We acknowledge all study participants, study community members, research team, APHRC staff and ARISE consortium members for their guidance and input during this project.

**Conflicts of Interest:** The authors have no conflict of interest to declare.

## Abbreviations

| | |
|---|---|
| APHRC | African Population and Health Research Center |
| ARISE Hub | Accountability and Responsiveness in Informal Settlements for Equity |
| CAC | Community advisory committee |
| CHH | Child-headed households |
| COREQ | Criteria for reporting qualitative research |
| CPBR | Community-based participatory research |
| ESRC | Ethics and Scientific Review Committee |
| IDIs | In-depth interviews |
| LSTM | Liverpool School of Tropical Medicine |
| MVGs | Marginalized and vulnerable groups |
| NACOSTI | National Commission for Science, Technology and Innovation |
| NCSS | Nairobi Cross-Sectional Slum Survey |
| NUHDSS | Nairobi Urban Health and Demographic Surveillance System |
| PWD | Persons with disability |
| SWM | Solid-waste management |

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
