# Peer review of "Unmet Needs and Resilience: The Case of Vulnerable and Marginalized Populations in Nairobi’s Informal Settlements"

_sustainability, doi:10.3390/su15010037_

Round 1
Reviewer 1 Report
Thank you very much for the opportunity to read the research paper. After reading the presented paper several questions may be raised. First, whether it answers the scope of the journal Sustainability and the aims of the Special issue „Promoting and Sustaining Urban Health: Challenges and Responses“. Second, whether introduces the research is scientifically justified and/or can be improved within the same frame.
The paper presents research without a focused relation with the aims of the journal Sustainability and the aims of the Special issue „Promoting and Sustaining Urban Health: Challenges and Responses“. The paper requires a more focused scientific attention to them. Also it might be considered to expand in the chapters of the literature review, research results, conclusions, discussion and the further research on the relatedness of the paper with the aims and scope of the journal and a special issue and to resubmit.
The authors might wish to consider the following issues for improvement:
Abstract:
- In the abstract, the objective of the research, research gap, research method and scope as well as major findings and practical implications should be presented.
Introduction:
- the terminology is not explained and justified, only abbreviations. They should refer to some scientiifc or official sources and also be explained.
- the scientific research gap, the objective and the aims of the research are not clear.
- The elements of scientific novelty are not clear. The "Introduction" chapter shows some of relevance of the current situation, though the scientific significance and what remains unsolved is not revealed.
- The structure of the publication is to be explained in the end of the Introduction.
Literature review:
- The presented paper doesn‘t have a section for scientific literature review and the analysis of scientific literature is not revealed in the paper anywhere. Therefore the current status of the research is not presented.
- Though the authors go straight to the conceptual framework but here they present Maslow hierarchy pyramid without analytical integration with the issues raised in this particular paper. Authors‘ scientific contribution is not visible which is expected in the conceptual framework.
Methodology:
- The research design and protocol, research ethics, data privacy issues, inclusion and exclusion criteria to justify the research scope should be explained.
- research ethics is not only about having written agreemnt, it is also about the researchers‘ role in the interviews, ensuring that there is not pressure on respondents to express their opinion, translations from – to languages, translation from visual communication to data for the research, etc.
- The research methods are mentioned. The process of the research is not explained and the combination of the methods is not justified. A methodological explanation is required to understand the design of the research and to prove the choice.
- The research framework including the criteria for the selection of the respondents as well as research data analysis is not presented and justified as an outcome of the scientific literature review.
- the selection of the respondents as well as the criteria to identify reasonable respondents is not explained
- the period of the is not indicated, consistency of the language translation is not explained
- one way of the research data collection is photos. It is not explained how data from visual pictures are translated into the data for the research
- Themes for Analysis (table 2) is a replication of Maslow‘s hierarchy pyramid. Content analysis is mentioned for the reading the transcripts, though data interpretation doesn‘t meet the usual presentation of the results.
- results should include only results and theoretical part (e.g. 293-298 and similar places) should be in the literature review part.
- 24 respondents should be coded and the citations of 24 respondents should be visible. Especially keeping in mind that there were 4 different methods sued for the research with all 24 respondents.
- four types of methods are introduced to collect data. The results of each of them and evolution of the results could be presented.
Discussion:
- Discussion chapter might be improved by showing a debate between this research results and recent research scientific publications.
Conclusions:
- Although the authors present the added value of this paper, the authors might wish to consider explaining the practical and academic implications.
- A future research agenda needs to be provided in the concluding remarks.
List of references:
- There are numerous amendments needed in the references. In many places the publishing house is missing, in some places (e.g. 14) some other information is missing. Links should work from anywhere.
Author Response
Reviewer One
Thank you very much for your time and for your expertise.
|
|
|
|
Questions/ Comments |
How we addressed the comments |
|
After reading the presented paper several questions may be raised. First, whether it answers the scope of the journal Sustainability and the aims of the Special issue „Promoting and Sustaining Urban Health: Challenges and Responses“. Second, whether introduces the research is scientifically justified and/or can be improved within the same frame. The paper presents research without a focused relation with the aims of the journal Sustainability and the aims of the Special issue „Promoting and Sustaining Urban Health: Challenges and Responses“. The paper requires a more focused scientific attention to them. Also it might be considered to expand in the chapters of the literature review, research results, conclusions, discussion and the further research on the relatedness of the paper with the aims and scope of the journal and a special issue and to resubmit.
|
Literature review has been done to link sustainability and our study topic. This is because unmet needs are health and wellbeing needs while resilience strategies are components of sustainability. |
|
The authors might wish to consider the following issues for improvement: Abstract |
|
|
- In the abstract, the objective of the research, research gap, research method and scope as well as major findings and practical implications should be presented. |
This has been done (Page 2). Thanks |
|
Introduction: |
|
|
- the terminology is not explained and justified, only abbreviations. They should refer to some scientific or official sources and also be explained. |
‘unmet needs” terminology is described in (Page 3) |
|
- the scientific research gap, the objective and the aims of the research are not clear. |
A clarity has been provided. |
|
- The elements of scientific novelty are not clear. The "Introduction" chapter shows some of relevance of the current situation, though the scientific significance and what remains unsolved is not revealed. |
This has been done (Page 4). Thanks |
|
- The structure of the publication is to be explained in the end of the Introduction. |
This has been done (Page 4). Thanks |
|
Literature review: |
|
|
- The presented paper doesn‘t have a section for scientific literature review and the analysis of scientific literature is not revealed in the paper anywhere. Therefore the current status of the research is not presented. |
The literature has been embedded in the introduction section. |
|
- Though the authors go straight to the conceptual framework but here they present Maslow hierarchy pyramid without analytical integration with the issues raised in this particular paper. Authors‘ scientific contribution is not visible which is expected in the conceptual framework. |
The authors’ contribution is provided below the pyramid. |
|
Methodology: |
|
|
- The research design and protocol, research ethics, data privacy issues, inclusion and exclusion criteria to justify the research scope should be explained. |
The research design, ethics and inclusion criteria has been provided throughout at the methodology section. |
|
- research ethics is not only about having written agreemnt, it is also about the researchers‘ role in the interviews, ensuring that there is not pressure on respondents to express their opinion, translations from – to languages, translation from visual communication to data for the research, etc. |
An extension of ethical principles adhered to has been provided (Page 8-9) |
|
- The research methods are mentioned. The process of the research is not explained and the combination of the methods is not justified. A methodological explanation is required to understand the design of the research and to prove the choice. |
A justification for governance diaries has been provided (Page 5). In-depth interviews was the dominant data collection method and the other methods informed the governance diaries. |
|
- The research framework including the criteria for the selection of the respondents as well as research data analysis is not presented and justified as an outcome of the scientific literature review. |
This has now been justified (Page 6) |
|
- the selection of the respondents as well as the criteria to identify reasonable respondents is not explained |
This has now been justified (Page 6) |
|
- the period of the is not indicated, consistency of the language translation is not explained |
This has been explained (Page 8) |
|
- one way of the research data collection is photos. It is not explained how data from visual pictures are translated into the data for the research |
This has now been justified (Page 5-6). Where photos, participant diaries, informal discussions and reflective sessions would inform robust IDIs; that was the dominant data collection method to be analysed. |
|
- Themes for Analysis (table 2) is a replication of Maslow‘s hierarchy pyramid. Content analysis is mentioned for the reading the transcripts, though data interpretation doesn‘t meet the usual presentation of the results. |
This has been re-worked. Framework analysis was used. |
|
- results should include only results and theoretical part (e.g. 293-298 and similar places) should be in the literature review part. |
This has been factored |
|
- 24 respondents should be coded and the citations of 24 respondents should be visible. Especially keeping in mind that there were 4 different methods sued for the research with all 24 respondents. |
One dominant method was coded. The other methods informed the IDIs; coded method. |
|
- four types of methods are introduced to collect data. The results of each of them and evolution of the results could be presented. |
A governance diaries methods was used; with IDIs as the dominant method. |
|
Discussion: |
|
|
- Discussion chapter might be improved by showing a debate between this research results and recent research scientific publications. |
This has been actioned at the discussion section |
|
Conclusions: |
|
|
· Although the authors present the added value of this paper, the authors might wish to consider explaining the practical and academic implications. |
· |
|
· A future research agenda needs to be provided in the concluding remarks. |
· This has been provided (Page 17) |
|
List of references: |
|
|
- There are numerous amendments needed in the references. In many places the publishing house is missing, in some places (e.g. 14) some other information is missing. Links should work from anywhere. |
This has been actioned |
|
|
|
Reviewer 2 Report
This was a very clearly written, well structured and enjoyable paper to read. The application of the Maslow model was well justified and the links to the MVGs made this of high societal relevance and impact. The use of the governance diaries approach was novel and I particularly liked the role of the community advisors in this as a way of engaging and encouraging participation from groups who may otherwise have been under-represented. The triangulation of multiple data types gave the work robustness and there was clear potential for rich in-depth data to be provided in this project. I have a couple of areas however which I was less clear on and which I feel would benefit from further justification or consideration.
1) Although the rationale for looking at unmet needs from Maslow's perspective was well described, I felt that further consideration of other strengths-based approaches would have benefited the paper. This could have included differentiating this perspective from other similar perspectives and highlighting in more depth/detail about how Maslow's approach was the most suitable in the current context. This would allow the authors to demonstrate a better supported argument. However this is a minor point and would only require a little additional work.
2) Given the nature of the data being collected and the groups being approached to participate, I think a clearer account of what participants were told about the nature of the study is important. Work of this type can often involve imbalanced power relations - so did the participants have any "expectations" about what the results of this study would achieve? How were those expectations managed? For example was this part of the training of those supporting the implementation of the project on the ground and how does this map to ethical guidelines and principles? I think further information about the engagement of participants and more explicit links to established frameworks of ethics and conduct are important in such a study.
3) One thing which I was not clear on was the language in which the data were collected. While this might well involve multiple languages and dialects, the mechanisms for translation are discussed in a small section on the bottom of page 5 and the top of page 6. What approach was taken for dialects within the main language, how was the accuracy of translation completed and checked, were transcripts back translated for validity etc? Given that this is qualitative data, more depth/detail on how the text was established would be useful.
4) Given the richness of the data and the mechanisms used to ensure robustness and triangulation, I was surprised about the choice of content analysis as an analytic approach. This seemed to consign the results of the study to a more superficial set of descriptions; a perception that was confirmed on reading the results themselves. In fact, given the extensive engagement with methods, the results were somewhat disappointing in presentation and conclusion. It is perfectly acceptable to use content analysis but I would suggest that a stronger justification for the use of this rather than even something like reflexive thematic analysis (Braun and Clarke) is necessary to convince the audience - including those who may be implementing policy and practice. I would suggest a section in or just before the data analysis section to present this justification.
5) Related to this was the presentation of results - which, although organised by the relevant headings, was the least logically coherent section of the paper. I found that even for a content analysis this was a somewhat descriptive account rather than an analytic one and the discussion of the themes and quotes was overshadowed by the quotes themselves. I feel that this needs to be more analytic, to have a much stronger argument contained within it and to redress the balance between narrative and quotation in favour of a stronger narrative in which the quotes "work to validate" the discussion of the categories. I felt that this was particularly important as the Maslow hierarchy had been imposed on this analysis. It was therefore not just crucial to see "what" the needs were but also what psychological meaning these had for the participants.
I think that this is an important and relevant study. However for me the presentation of results undersells the potential impact of the work and is confusing for the reader. Therefore addressing the questions around the analytic approach and results presentation would be the most important (points 4 & 5) and then the others can be dealt with as more minor points during the revision process.
Author Response
Reviewer Two
|
Questions/ Comments |
How we addressed the comments |
|
This was a very clearly written, well structured and enjoyable paper to read. The application of the Maslow model was well justified and the links to the MVGs made this of high societal relevance and impact. The use of the governance diaries approach was novel and I particularly liked the role of the community advisors in this as a way of engaging and encouraging participation from groups who may otherwise have been under-represented. The triangulation of multiple data types gave the work robustness and there was clear potential for rich in-depth data to be provided in this project. I have a couple of areas however which I was less clear on and which I feel would benefit from further justification or consideration. |
Thank you. We acknowledge your time and expertise. |
|
1) Although the rationale for looking at unmet needs from Maslow's perspective was well described, I felt that further consideration of other strengths-based approaches would have benefited the paper. This could have included differentiating this perspective from other similar perspectives and highlighting in more depth/detail about how Maslow's approach was the most suitable in the current context. This would allow the authors to demonstrate a better supported argument. However this is a minor point and would only require a little additional work. |
We did not include other strength based methods because to the best of our knowledge Maslows hierarchy of needs, bests categorizes human needs in order of priority- stating how lower level needs cannot be fully satisfied before seeking for higher level needs. |
|
2) Given the nature of the data being collected and the groups being approached to participate, I think a clearer account of what participants were told about the nature of the study is important. Work of this type can often involve imbalanced power relations - so did the participants have any "expectations" about what the results of this study would achieve? How were those expectations managed? For example was this part of the training of those supporting the implementation of the project on the ground and how does this map to ethical guidelines and principles? I think further information about the engagement of participants and more explicit links to established frameworks of ethics and conduct are important in such a study. |
Participants’ expectations were managed from the onset and power imbalances were managed from the fact that co-researchers from the community were trained to collect the data. Further, reflections and informal discussions enabled ease the power imbalance between researcher and co-researchers and between co-researchers and study participants. (Described in different sections of the methodology) |
|
3) One thing which I was not clear on was the language in which the data were collected. While this might well involve multiple languages and dialects, the mechanisms for translation are discussed in a small section on the bottom of page 5 and the top of page 6. What approach was taken for dialects within the main language, how was the accuracy of translation completed and checked, were transcripts back translated for validity etc? Given that this is qualitative data, more depth/detail on how the text was established would be useful. |
We collected the data using Swahili translated tool. We ensured accuracy of data by reading through transcripts multiple times and cross-checking. (Described in the data collection section) |
|
4) Given the richness of the data and the mechanisms used to ensure robustness and triangulation, I was surprised about the choice of content analysis as an analytic approach. This seemed to consign the results of the study to a more superficial set of descriptions; a perception that was confirmed on reading the results themselves. In fact, given the extensive engagement with methods, the results were somewhat disappointing in presentation and conclusion. It is perfectly acceptable to use content analysis but I would suggest that a stronger justification for the use of this rather than even something like reflexive thematic analysis (Braun and Clarke) is necessary to convince the audience - including those who may be implementing policy and practice. I would suggest a section in or just before the data analysis section to present this justification. |
Framework analysis and its justification has been described. (Described in the Methodology section).
The results section has been reworked |
|
5) Related to this was the presentation of results - which, although organised by the relevant headings, was the least logically coherent section of the paper. I found that even for a content analysis this was a somewhat descriptive account rather than an analytic one and the discussion of the themes and quotes was overshadowed by the quotes themselves. I feel that this needs to be more analytic, to have a much stronger argument contained within it and to redress the balance between narrative and quotation in favour of a stronger narrative in which the quotes "work to validate" the discussion of the categories. I felt that this was particularly important as the Maslow hierarchy had been imposed on this analysis. It was therefore not just crucial to see "what" the needs were but also what psychological meaning these had for the participants. |
This has been done with implications provided for the needs. Thanks The results section was reworked. |
|
I think that this is an important and relevant study. However for me the presentation of results undersells the potential impact of the work and is confusing for the reader. Therefore addressing the questions around the analytic approach and results presentation would be the most important (points 4 & 5) and then the others can be dealt with as more minor points during the revision process. |
This has been done. Thanks |
Reviewer 3 Report
In principle the study and article are well described and written. Also, theoretical background is sufficiently described. The main flaw in the article is the absence of criticism to the result and of the theory (e.g. theory claims that "When physiological needs are relatively satisfied, new needs arise that are classified as safety needs (35). " but "There were no self-actualization needs mentioned. However, it does not need full satisfaction to be able to move from a certain need to a higher one, as people who are satisfied with some upper-level needs may sometimes still have lower-level needs." Similar conflicts are mentioned elsewhere also but can be explained by resilience strategies partly. Unmet needs that have been identified in the study correlate well with other data, but this makes the results also quite obvious, so I would wish that authors could have raised more the value of the study still. Despite these comments, I think that the study has been valuable work and could be further developed to be even more useful.
Author Response
Reviewer Three
|
Questions/ Comments |
How we addressed the comments |
|
In principle the study and article are well described and written. Also, theoretical background is sufficiently described. The main flaw in the article is the absence of criticism to the result and of the theory (e.g. theory claims that "When physiological needs are relatively satisfied, new needs arise that are classified as safety needs (35). " but "There were no self actualization needs mentioned. However, it does not need full satisfaction to be able to move from a certain need to a higher one, as people who are satisfied with some upper level needs may sometimes still have lower level needs." Similar conflicts are mentioned elsewhere also but can be explained by resilience strategies partly. Unmet needs that have been identified in the study correlate well with other data, but this makes the results also quite obvious, so I would wish that authors could have raised more the value of the study still. Despite these comments, I think that the study has been valuable work and could be further developed to be even more useful. |
The top higher level need is always impossible to achieve when lower level needs are not met to some extent. Page 4 |
Reviewer 4 Report
This research focuses on the Unmet needs and resilience of vulnerable and marginalized populations; comments below could help improve the paper's overall structure and writing.
- Defining the research question, the research goal, and how it reflects the study's findings. First, the research context is of great importance ( informal settlements). Accordingly, it would help if you rewrote the question by adding this context: what are the unmet needs of MVGs in informal settlements) ?
- Another point is that your research deals with priorities of the needs from the citizen's lens. It means that the goal of your study is a bit different, and you should clarify it at the beginning of the paper and be more consistent throughout the overall article.
- Defining research question – you also ask, "what are the resilient strategies embraced by MVGs in two informal settlements in Nairobi, Kenya? Should we learn about resilient strategies in informal settlements in general or about two specific locations? It seems that you are having difficulty To Generalize your findings. It brings me to another point that is pretty missing here- what is the main contribution of your study? How and whether can scholars learn from those two cases? Please provide us a rationale for choosing these two localities and the main differences and similarities between Korogocho and Viwandani in terms of Size, etc. in methodology section. How many informal settlements are in the country, and why this phenomenon exists?
There is a problem with consistency in times in your writing. It would help if you wrote about the study in the present time, while your methods should be wroten in the past. For example: – not the respondents should do.. but respondents did// answered .. for instance, you write:" Each participant would write (the study already has been accomplished so they wrote ) about daily activities related to unmet needs 169 and resilient strategies.
- Please explain why the governance diaries method is suitable and appropriate for this specific study and how it assists in promoting its goal.
- You write that you provided the study participants with guidelines – which guidelines? Please present the examples.
- Figure 3- should match the explanation and description you provided in the data collection section, but there are some mismatches. What about the pictures- you should remove them.
- Please provide the advantages of NVivo software for your research while you describe, in short, what this software is
- Inaccuracy in the table - The themes you presented in the table are not emerging but were defined by you in advance, as you wrote at the beginning of the paper.
- It would help if you classified the respondents' answers, for example, R1 (from Settlement X or Y)
- You can not start a discussion with the sentence: The study findings will contribute to policy and practice. First, please elaborate on why and how, referring to my previous comment on the main contribution. Second, you should emphasize it in conclusion.
- Cognitive strategies – you mention that The cognitive coping strategies included acceptance,- acceptance is a tricky element as acceptance could result from indifference. Please explain more deeply how it relates to resilience and why.
- Proofreading, highlighting, and reorganizing the titles are required.
Author Response
Reviewer Four
|
Questions/ Comments |
How we addressed the comments |
|
This research focuses on the Unmet needs and resilience of vulnerable and marginalized populations; comments below could help improve the paper's overall structure and writing. |
|
|
The research questions have been re-casted accordingly. Thank you for the suggestion |
|
This has been clarified. Thanks |
|
Research questions have been amended accordingly |
|
There is a problem with consistency in times in your writing. It would help if you wrote about the study in the present time, while your methods should be wroten in the past. For example: – not the respondents should do.. but respondents did// answered .. for instance, you write:" Each participant would write (the study already has been accomplished so they wrote ) about daily activities related to unmet needs 169 and resilient strategies. |
This has been amended accordingly. Thanks. |
|
This has been described at the methods sections |
|
The participants were presented with guidelines on how to document the participant diaries. This has bee described further. |
|
The text has been re-organized to be in tandem with the data collection process. Thanks |
|
This has been provided. Thanks |
|
This has been re-worked on accordingly as they are sub-themes. Thanks |
|
|
|
This has been reworked on accordingly. Thanks |
|
Further clarification has been provided. Thanks. |
|
This has been done. Thanks |
Round 2
Reviewer 1 Report
Despite the authors' claim that they made all corrections, there are significant comments that are not taken into account. Some of them:
- Many terms are still not explained even if unmet needs is explained.
- The link between the research and sustainability is not scientifically justified. To have a sound background the paper might require restructuring the literature review part. It looks like sustainability is used as a term understood by default by everyone but this link is not so obvious scientifically.
- there are amendments in the methodological part but still there is no clear structure of the research design. Therefore it looks like a mix of different methods without clarity of how the inputs or outputs of the research of each data collection method supplements each other. Also, a justification based on previous research would be important to show the validity of the design.
- The Maslow Pyramid presented as a conceptual framework, in reality, has no authors‘ input, it is a visualization of Maslow‘s pyramid. The source of the pyramid (or the framework) is not introduced thus suggesting authors of the paper as authors of the pyramid.
- Maslow's pyramid, sustainability, and research topic are not interlinked or it is not clearly visualized.
- The list of literature is not improved and necessary information is not provided in many pieces of the reference.
The paper even if the research done could look promising is still not ready to be published in the journal in Q2 and with a high IF. The paper should be improved and restructured or maybe resubmitted.
Author Response
Thank you for taking your time and for your expertise. Kindly find the responses in italics
1. Many terms are still not explained even if unmet needs is explained.
The terms have already been defined. Thanks
2. The link between the research and sustainability is not scientifically justified. To have a sound background the paper might require restructuring the literature review part. It looks like sustainability is used as a term understood by default by everyone but this link is not so obvious scientifically.\
The research had been linked to SDG 11 on sustainable cities. Thanks
3. There are amendments in the methodological part but still there is no clear structure of the research design. Therefore it looks like a mix of different methods without clarity of how the inputs or outputs of the research of each data collection method supplements each other. Also, a justification based on previous research would be important to show the validity of the design.
This has been clarified. Thanks
4. The Maslow Pyramid presented as a conceptual framework, in reality, has no authors‘ input, it is a visualization of Maslow‘s pyramid. The source of the pyramid (or the framework) is not introduced thus suggesting authors of the paper as authors of the pyramid.
The Figure has been revised accordingly. Thanks
5. Maslow's pyramid, sustainability, and research topic are not interlinked or it is not clearly visualized.
The Figure has been revised accordingly. Thanks
6. The list of literature is not improved and necessary information is not provided in many pieces of the reference.
Literature section has been improved
7. The paper even if the research done could look promising is still not ready to be published in the journal in Q2 and with a high IF. The paper should be improved and restructured or maybe resubmitted
The paper has been restructured. Thanks
Once again thanks you.

Reviewer 2 Report
I am happy that my recommendations have been addressed. The method and results section in particular are much improved and I am happy that this meets my requirements.
Author Response
I am happy that my recommendations have been addressed. The method and results section in particular are much improved and I am happy that this meets my requirements.
Thanks
Reviewer 3 Report
55. Resilience B. ...because ...because our our shared shared vision of tomorrow,. 2021;
Above, odd reference.
The answer the authors gave is not fully satisfactory as basically they explain that resilience strategy can change the order or importance of the needs. I would like to see some critical view to the Maslow model too, at least from this point. You can easily find critical commentary about Maslow's hierarchy, and I think it would be important to refer to those at some level, Now the model is taken as "divine announcement",
Author Response
Thank you for your time and expertise. Kindly find the responses in italics.
- Resilience B. ...because ...because our our shared shared vision of tomorrow,. 2021;
The references have been reviewed
Above, odd reference.
The answer the authors gave is not fully satisfactory as basically they explain that resilience strategy can change the order or importance of the needs. I would like to see some critical view to the Maslow model too, at least from this point. You can easily find critical commentary about Maslow's hierarchy, and I think it would be important to refer to those at some level, Now the model is taken as "divine announcement",
Acritical commentary has been provided
Once again thanks.

Round 3
Reviewer 1 Report
Despite space for advancement still left, the publication has been improved.
Author Response
|
S/n |
Comment |
How we addressed |
Line |
|
1 |
Title: Make the tile more appealing |
The Title has been adjusted accordingly |
2-3 |
|
2 |
The sentence is incomplete |
The words that made the sentence to look incomplete have been deleted |
24-25 |
|
3 |
SDG 3 has not been mentioned throughout the paper |
SDG 3 correlation with SDG has been mentioned. Further SDG 3 has been mentioned at the introduction, results and discussions |
68-69; 72-73 (had already been mentioned); 253-363 |
|
4 |
Details on population of residents in the study sites |
This has been done accordingly |
212-214 |
|
5 |
The need to acknowledge that it is an exploratory study, since there were only 24 study participants |
This had already been done at the limitation section |
635 |
|
7 |
Recast the following: Health needs
|
Changed to Health care and medical service needs
|
353 |
|
8 |
References |
Incite citations and references have been reworked accordingly |
Throughout the paper |
|
9 |
Semantic errors and suggestions |
Semantic errors and suggestions have been adjusted accordingly |
Throughout the paper |